# Titanium Lattice Structures Produced via Additive Manufacturing for a Bone Scaffold: A Review

**DOI:** 10.3390/jfb14030125

**Published:** 2023-02-24

**Authors:** Fabio Distefano, Salvatore Pasta, Gabriella Epasto

**Affiliations:** 1Department of Engineering, University of Messina, C.da Di Dio, 98166 Messina, Italy; 2Department of Engineering, University of Palermo, Viale delle Scienze, 90128 Palermo, Italy; 3Department of Research, IRCCS ISMETT, Via Tricomi n.5, 90127 Palermo, Italy

**Keywords:** lattice structures, titanium alloy, bone tissue engineering, scaffolds, additive manufacturing, mechanical properties

## Abstract

The progress in additive manufacturing has remarkably increased the application of lattice materials in the biomedical field for the fabrication of scaffolds used as bone substitutes. Ti6Al4V alloy is widely adopted for bone implant application as it combines both biological and mechanical properties. Recent breakthroughs in biomaterials and tissue engineering have allowed the regeneration of massive bone defects, which require external intervention to be bridged. However, the repair of such critical bone defects remains a challenge. The present review collected the most significant findings in the literature of the last ten years on Ti6Al4V porous scaffolds to provide a comprehensive summary of the mechanical and morphological requirements for the osteointegration process. Particular attention was given on the effects of pore size, surface roughness and the elastic modulus on bone scaffold performances. The application of the Gibson–Ashby model allowed for a comparison of the mechanical performance of the lattice materials with that of human bone. This allows for an evaluation of the suitability of different lattice materials for biomedical applications.

## 1. Introduction

Lattice structures are topologically ordered structures based on one or more repeating unit cells [1]. From Gibson–Ashby’s research on cellular solids, a unit cell is determined by the connectivity and dimensions of its constitutive strut elements, which are connected at specific nodes [2]. Lattice materials present voids deliberately embedded in their structures [2]. Three types of lattice structures are currently studied for engineering applications: strut-based, triply periodic minimal surfaces (TPMS) skeletal and TPMS sheet [3].

In recent years, the application of lattice structures in engineering fields has considerably grown due to the progress in Additive Manufacturing (AM). Freedom of design, mass customization, waste minimization, the ability to manufacture complex structures and rapid prototyping are the major advantages of AM [4]. The ability to fabricate components with complex parts and customizable material properties is one of the most important advantages of these technologies, allowing for the production of complex functional objects from multiple materials unattainable with conventional manufacturing methods [5]. The modern approaches to fabricate bone constructs via AM provide a favourable environment for bone regeneration [6]. Different AM technologies are currently used for the fabrication of parts from metallic fine powders, for instance: Selective Laser Melting (SLM) [7], Selective Laser Sintering (SLS) [8], Direct Metal Laser Sintering (DMLS) [9], Electron Beam Melting (EBM) [10].

Lattice materials attract a great deal of interest in several engineering disciplines, for instance automotive and aerospace, thanks to the high strength-to-weight ratio, thermal conductivity and enhanced mechanical energy absorption [11,12]. Applications in the biomechanical field used as bone substitutes have been also proposed [13,14,15,16].

The Ti6Al4V alloy has a long history of application for bone implants due to its mechanical biocompatibility, high strength, long lifetime, high wear resistance and low elastic modulus [17,18,19].

The mechanical and morphological features of lattice materials affect the osteointegration and bone ingrowth of the candidate implant. Specifically, the elastic modulus was demonstrated to affect scaffold remodelling [20], as well as cell migration and differentiation [21]. It is particularly important to reduce or eliminate stress shielding, which is one of the primary causes requiring revision surgery leading to bone resorption [22,23]. Other important factors are pore size, which affects cell penetration and bone ingrowth [24,25], and surface roughness, which supports the achievement of improved interaction mechanisms between the implant and biological tissues [26,27].

This review aims to provide a comprehensive view of the mechanical and morphological requirements of lattice structures for the design of biomedical implants for bone substitutes, with a focus on the effect of pore size and surface roughness on the bone ingrowth and on the effect of the elastic modulus in the reduction of stress shielding and promotion of osseointegration. The current work intends to furnish guidelines on the choice of the most suitable lattice topology by applying the Gibson–Ashby model and by focusing on the performance and features needed for bone stimulation in the osseointegration process. The review has collected the most significant findings achieved in the last ten years with the utilization of additive manufacturing with Ti6Al4V for porous scaffolds.

## 2. Classification of Lattice Structures

Three classes of lattice structures are commonly investigated in applications for biomedical engineering: the first class is strut-based structures, while the second and third classes are derived from mathematically-created TPMS, namely skeletal-TPMS and sheet-TPMS lattice structures. 

Figure 1 shows the strut-based unit cell [2,28] proposed by Gibson–Ashby.

The most known strut-based topologies, which are named after analogous crystalline structures, are the body centred cubic (BCC) and face centred-cubic (FCC), as well as the variations named z-struts BCCZ and FCCZ. There exist other strut-based topologies, such as the cubic, octet-truss and diamond (see Figure 2).

The octet-truss cell is composed of an octahedral cell (black part) and a tetrahedral cell (light grey part), as shown in Figure 3.

Other variations in the BCC lattice are represented by the G7 unit cell and the simple cubic body-centred cubic (SCBCC) as shown in Figure 4.

Lattice materials with cubic symmetry, strut-based cell topology, such as Archime-dean solids or Catalan solids, have been widely investigated over the years. Archimedean solids are a group of 13 solids, first enumerated by Archimedes (see Figure 5). They are convex uniform polyhedra composed of regular polygons with identical vertices. Catalan solids are the dual polyhedra of the Archimedean solids. Most studied Archimedean solids include the cuboctahedron [32,33], truncated cube (TC) [34,35], truncated octahedron (TO or Kelvin cell) [36,37], rhombicuboctahedron (RCO) [35,38] and truncated cuboctahedron (TCO) [35,39]. Among Catalan solids, researchers mostly focused their attention on the rhombic dodecahedron (RD) [35,40]. 

Recently, cellular structures with mathematically defined architectures, as TPMS based topologies, have been proposed [3]. A minimal surface can be considered as a surface with a mean curvature equal to zero in all points; thus, a TPMS is characterized as a minimal surface periodic in three independent directions [41]. TPMS mathematical representation is defined by a system of coordinates calculated using the Enneper–Weierstrass parametric representation as shown here.
(1){x=Re(eiθ∫w0w(1-τ2)R(τ)dτ)y=Re(eiθ∫w0wi(1-τ2)R(τ)dτ)z=Re(eiθ∫w0w2τR(τ)dτ)

R(τ) represents a function dependent upon the TPMS topology and can be expressed as:(2)$$R\left(t\right)=\frac{1}{\sqrt{{1–14\tau }^{4}{+\tau }^{8}}}$$

Compared to the parametric form, a TPMS has a simpler and unified representation expressed by sinusoid terms and defined as [41]:(3)$$\varphi \left(\gamma \right)=\sum_{k=1}^{K}A_{k}\cos\left[\frac{2\pi \left(h_{k}\cdot \gamma \right)}{\lambda_{k}}{+p}_{k}\right]=C$$

TPMS topologies can be approximately defined as combinations of trigonometric functions in an implicit form. Examples of the most common TPMS equations in the implicit form are expressed in Table 1 as follows [42,43]:

Two classes of TPMS lattices can be considered, namely sheet TPMS and skeletal TPMS. Some TPMS lattices were investigated in both versions, such as gyroid [44,45], diamond [15,46] and IWP [47,48]. Other TPMS structures have been studied only in the sheet-based version: Schwarz primitive [49,50], FRD [47,51] and Neovius [52,53]. Figure 6 shows commonly known TPMS structures.

Over the last few years, novel lattice materials have been proposed (see Figure 7). Dong et al. [54] numerically and experimentally investigated the mechanical behaviour of a vintile single unit cell and lattice, while other researchers analysed its behaviour for biomedical applications [55,56]. Alomar et al. [57] proposed and developed a new lattice material based on a circular constituent cell. Distefano et al. [58] developed a novel biomimetic lattice material based on the scheme of rocks and called the TAOR lattice.

## 3. Current Status of Additive Manufacturing Technologies

AM has grown considerably in recent years, thanks to the technological advancement and the subsequent enhanced material properties. The ability to create components with complex parts and customisable material properties is one of the most important advantages of AM, allowing the fabrication of complex functional objects from various materials unattainable with conventional manufacturing methods [5]. This resulted in the industrial use of AM parts, even in highly advanced applications, most notably in aerospace, automotive and biomedical fields. Different AM technologies are currently used for the fabrication of parts used in this fields, from metallic fine powders, for instance: SLM, DMLS, EBM [10].

However, due to the rapid diffusion of a multitude of technologies related to AM, there is a lack of a comprehensive set of design principles, manufacturing guidelines and standardization of best practices. AM techniques require process optimization and quality control to ensure accuracy and reliability [60]. This requirement is particularly relevant for components with complex geometries, such as lattice structures, which include curved surfaces and thin connecting features. Different factors, such as machine selection, processes and materials, position and orientation of the part and finishing can alter the resulting quality of the printed component [61]. A major limitation is the minimum feature size for the AM technology used [62], the achievable feature resolution is inherently constrained by the fact that powder-based technologies require particles larger than 20 μm so that the powder can be successfully spread during recoating [63]. An additional limitation is placed on the part design, most notably the build angles [64]. When extremely complex structure, such as truncated icosahedra, are printed with dimensions in the order of micrometres, some feature cannot be reproduced [65]. An important attribute is the surface quality, which is mainly determined by the thickness of each printed layer. Surface quality also depends on the form of the raw material; powder bed AM processes present poorer surface quality than others due to large and partially melted powder particles that reside on the printed part’s surface [61].

In order to promote research interest and investments, the goal of AM technologies is to face these and other challenges to ensure the quality of the 3D-printed products [66].

## 4. Mechanical and Morphological Requirements for Biomedical Applications

Bone is a complex tissue undergoing biological remodelling. This feature of bone underpins the ability to remodel itself to repair damage [67]. However, the bone’s ability for self-regeneration of massive defects can be limited because of deficiencies in blood supply or in the presence of systemic disease [68]. When bone defects exceed a critical non-healable size, a surgical process is required to support self-healing when defects need bridging. Despite recent advances in biomaterials and tissue engineering, the repair of such a critical bone defects remains a challenge [69,70,71]. Lattice materials are widely used in the biomedical field when devices are applied as bone substitutes [72,73,74,75,76,77,78,79]. A successful porous metallic implant should restore the physiological function of the bone, and porous structures provide high interfacial bond area for vascularization and bone ingrowth, promoting the biological fixation of implants and bone [80]. An ideal bone porous implant should possess the following properties: biocompatibility, suitable surface for cell attachment, proliferation, differentiation and migration [67]. In addition, angiogenesis is an essential physiological process for bone regeneration [81]; the biological inertia of the Ti6Al4V surface and the deficit of angiogenesis may cause postoperative complications, such as dislocation or loosening of the device [82]. Several methods to enhance the angiogenesis of Ti6Al4V scaffolds were evaluated in the literature, including the development of multifunctional surface coatings with angiogenic properties [83]; the incorporation into a Ti6Al4V alloy of copper ions, which presents high bioactivity and outstanding antibacterial properties [84]; and the controlled release of bone-morphogenic protein-2 and vascular endothelial growth factor in the Ti6Al4V alloy [85]. The abilities of bone-lining cells are affected by the size and shape of the scaffold pores; higher porosity promotes cell ingrowth and the transport of nutrients [86]. The current review presents a comparison of literature findings to evaluate the effect of mechanical and morphological properties on bone ingrowth and osseointegration and estimate the optimal parameters of lattice structures for biomedical applications.

### 4.1. Effect of Pore Size

The influence of pore size is still controversial. Wang et al. [67] reported that the optimal pore size is in the range of 100–400 μm. However, several studies evaluated the influence of pore size on bone ingrowth, with controversial results. Van Bael et al. [87] produced, via SLM, six Ti6Al4V scaffold configurations with three different pore shapes (triangular, hexagonal and rectangular) and two distinct pore sizes (500 μm and 1000 μm); after 14 days, in vitro culture exhibited a significantly higher living cell density on the Ti6Al4V bone scaffolds with 1000 μm pores. Cheng et al. [88] used a human trabecular bone template to design and manufacture Ti6Al4V scaffolds with variable porosities via SLS and observed increasing bone ingrowth up to 300 μm. The latter was considered an optimal value for bone ingrowth. Prananingrum et al. [89] fabricated porous titanium scaffolds stratified in four groups with increasing pore sizes from 60 μm up to 600 μm. After 20 weeks, the group with a pore size of 100 μm showed considerably greater bone ingrowth compared to the other groups. Li et al. [90] fabricated Ti6Al4V scaffolds with three pore sizes between 300 μm and 700 μm using Electron Beam Melting (EBM). Bone ingrowth in the group with a pore size of 400 μm was significantly higher than that with the other two porous scaffolds. Taniguchi et al. [91] implanted 300 μm, 600 μm and 900 μm porous titanium scaffolds into rabbit tibia and found that 600 μm and 900 μm scaffolds demonstrated significantly higher bone ingrowth than 300 μm scaffolds. Kapat et al. [92] used Ti6Al4V samples with 92 μm, 178 μm and 297 μm pore sizes; a quantitative evaluation of bone ingrowth via μ-CT revealed that there was approximately 52% higher bone formation in the sample with a 178 μm pore size after 12 weeks, compared to that with the other configurations. Ran et al. [93] designed and fabricated porous Ti6Al4V implants with straightforward pore dimensions: 500, 700, and 900 μm using SLM. They assessed the morphological features of scaffolds, showing that actual pore sizes were about 400, 600 and 800 μm; they observed that the biological performance of specimens with a 600 μm pore size was superior to that of the other two groups. Luan et al. [94] examined porous Ti6Al4V scaffolds with 334.1 μm, 383 μm and 401 μm pore sizes; results highlighted that all three types of porous Ti6Al4V scaffolds were inclined to promote bone ingrowth; however, a pore size of 383 μm showed better results. Ouyang et al. [95] fabricated, via SLM, porous titanium scaffolds with similar porosity and different pore sizes: 400, 650, 850 and 1100 μm; the best bone ingrowth was observed in scaffold with a 650 μm pore size. Chen et al. [96] manufactured, via SLM, scaffolds with 500 μm, 600 μm and 700 μm pore sizes and 60% and 70% porosities to explore the optimal morphological features. The scaffold with a pore size of 500 μm and porosity of 60% exhibited the best bone ingrowth by means of in vivo experiments. Wang et al. fabricated, via EBM, seven groups of porous scaffolds with pore sizes from 800 μm to 1000 μm. Bone ingrowth was assessed via μ-CT 3D reconstruction images showed the magnitude of positive remodelling with 1000 μm-pore-size scaffolds. Table 2 and Figure 8 describe the aforementioned findings in chronological order.

All presented results were obtained by performing in vivo or in vitro experiments where μ-CT reconstructions were applied to quantitatively show the effect of pore size on bone ingrowth (see Figure 9).

In brief, porosity, pore size and pore interconnectivity are important factors influencing the mechanical and biological properties of scaffolds, bone ingrowth and the transportation of cells and nutrients. Given the discrepancy among findings, the optimal topological scaffold architecture remains a major challenge in biomedical applications.

### 4.2. Effect of Surface Roughness

Another factor affecting bone ingrowth is the surface roughness. Porous samples with highly rough surfaces, Ra ≥ 56.9 μm, resulted in a reduction in proliferation and bone ingrowth [98], while the surface roughness with range of 0.5 up to 8.5 μm positively influenced the bone implant [26]. Surface roughness affects the permeability of porous implants. As surface roughness increases, the permeability decreases. High values of permeability enhance the osteointegration process, since the transportation of cells, nutrients and growth factors requires the flow of blood through the porous scaffolds [99]. Chen et al. [100] manufactured Ti6Al4V discs with different additive angles, via an SLM process, with the aim of maximizing the direct effects of the additive angle on surface properties and biocompatibility. As the angle increases, the surface roughness increases because of the increasement of unmelted metallic particles. They, in vitro, evaluated the effect to osteoblast attachment and proliferation with six surface roughness values ranging from 2 to 3 μm, in comparison to those with wrought samples (see Figure 10).

Adhesion and proliferation were found to be similar on the surface for every angle (see Figure 11), though cells initially adhered less with improved cell spreading at a higher additive angle.

Li et al. [101] prepared porous Ti6Al4V scaffolds via EBM; then, scaffolds were subjected to solution treatment at 800 °C, 950 °C and 1000 °C and then water quenching. Heat treatment increased surface roughness, obtaining values in the range of 3 up to 8 μm (see Figure 12).

The result showed that the scaffold heat treated at 1000 °C exhibited the best cellular adhesion and proliferation after in vitro culture (see Figure 13).

### 4.3. Effect of Elastic Modulus

Implants made of metals and alloys are usually stiffer than human bone mechanical properties; specifically, the Ti6Al4V alloy has an elastic modulus of nearly 110 GPa [102,103]. Mechanical properties of bone vary significantly with age, bone quality and the presence of diseases [104]. The elastic modulus continues to cause a scientific challenge to fully understand the mechanics of living bones [67]. Cortical bone is stiffer when loaded longitudinally than in transverse and shear directions and presents a longitudinal modulus ranging from 18 to 20 GPa, transverse modulus between 10 and 12 GPa and shear modulus of around 3 GPa [105,106,107,108]. Cancellous bone presents a low elastic modulus, ranging from 0.2 to 4 GPa [105,108,109,110]. This mismatch in the elastic modulus may lead to stress shielding, which represents a major issue for bone resorption and eventual failure of the implants [19,111]. The internal lattice architecture, i.e., porosity, pore size and pore interconnectivity, can be designed to lower the equivalent elastic modulus of the implant, avoiding the mismatch between the stiffness of the implant and the adjacent bone [112]. This will allow the matching of mechanical requirements of the bone substitute to reduce the stress shielding by satisfying loading requirements. This may however avoid the mechanical failure of the implant [113], by maintaining an appropriate mechanical strength. Indeed, porous scaffolds also have a load-bearing function [114,115].

The choice of the most suitable lattice topology that should be used for the conception of biomedical implants is still controversial. Parisien et al. [59] investigated the capability of strut-based lattices to enhance osteointegration. They compared twenty-four lattice topologies with ten different relative densities, from 5 up to 50%, subjected to bone ingrowth stimulations by applying four different pressures, in the range of 0.5 to 2 MPa. They evaluated the effect of the lattice elastic modulus on the percentage of pores that is optimal for bone ingrowth. Relative densities lower than 20% showed similar bone ingrowth performance. For relative densities higher than 30%, at the lowest pressure of 0.5 MPa, topologies with a smaller elastic modulus stimulate better bone ingrowth, while at the highest pressure of 2 MPa, the FCCZ, which has the highest elastic modulus, was the only topology showing less than 70% of bone stimulation (see Figure 14). Since various topologies presented more than 90% of their optimal space, the authors assessed that topology choice can be based on the elastic modulus that fits the design’s needs.

Several studies evaluated the effects of the elastic modulus of biomedical implants. Sun et al. [116] proposed a novel ZK60 cervical cage and evaluated the biomechanical properties under flexion, extension, lateral bending and axial rotation of the cervical spine. They performed cage optimization by decreasing the volume of 40% to reduce the cage’s stiffness. They observed that the optimized cage can considerably enhance the stress stimulation of the bone by reducing the stress-shielding effect between the implant and vertebral bodies. They also observed that the stresses at the endplate–cage interface decrease while the cage’s stiffness decreases, indicating that subsidence is less likely to occur in the device with lower stiffness. Wieding et al. [115] evaluated the mechanical performance of two types of Ti6Al4V-custom-made porous implants, with an elastic modulus of 6 and 8 GPa, implanted into a 20 mm segmental defect in the metatarsus of sheep. After 12 and 24 weeks postoperative, they performed torsional testing on the implanted bone and compared it to the contralateral non-treated side. Both types of implants offered mechanically stable situations, with bone tissue ingrowth around and into the implants, presenting an elastic modulus in the range of the cortical bone.

## 5. The Gibson–Ashby Model

### 5.1. Compressive Behaviour

Lattice structure can be categorized based on its mechanical response as bending-dominated or stretch-dominated. Bending-dominated structures are compliant and absorb energy well when compressed; stretch-dominated structures have greater stiffness and compressive strength compared to bending-dominated lattices, for a given relative density [117].

Figure 15 shows the compressive stress–strain curve of a bending-dominated lattice.

The deformation behaviour can be divided into three stages: linear elastic deformation, plastic deformation and densification. In the first stage, the lattice material response is linear elastic with a Young’s modulus proportional to the structure material compliance. Under compression, the struts of lattice materials are exposed to three mechanisms of collapse: yield, buckling or crushing. They compete until the mechanism with the lowest stress threshold is reached. Once the elastic limit is reached, plastic deformation begins and the structure keeps collapsing at a nearly constant stress, referred to as the plateau stress, until the opposite side of the cells impinge, constraining further deformation. The densification strain is reached, and densification begins as stress rises steeply [29,117].

Figure 16 shows the compressive stress–strain curve of a stretch-dominated lattice.

When subjected to a tensile/compressive loading, a stretch-dominated lattice material first responds via elastic stretch of the struts; on average, in the first stage of the curve only one-third of lattice strut bears loads [117]. In this case, the elastic limit is reached when one or more sets of struts yield plastically, buckle or crush. Once the elastic limit is reached, the whole lattice material bears the loads, and the structure fails through strut fracture. The stretch-dominated mechanisms of deformation involve hard modes (stretching) compared to the soft ones (bending) of the bending-dominated structures; therefore, initial yielding is followed by plastic buckling or brittle collapse of the struts, leading to post-yield softening, with oscillation of the stress required for further deformation. Then, the densification strain is reached, and the stress steeply increases.

The deformation behaviour of strut-based topologies can be estimated by Maxwell’s stability criterion, by evaluating the Maxwell number M of the lattice material.
(4)M=S - 3N+6

If M < 0, the structure is bending dominated; if M ≥ 0, the structure is stretch dominated; if M > 0 the structure is hyperstatic [118].

The Maxwell number of most commonly studied strut-based unit cells is reported in Table 3.

### 5.2. Comparison of Experimental Data for Different Lattice Materials

The Gibson–Ashby model is the most relevant and commonly recognized model for the prediction of the lattice’s mechanical properties, which depend on the deformation behaviour exhibited by the structure (bending or stretch-dominated) and show a positive power relationship with the structure relative density. Gibson–Ashby provided the formulae relating the elastic modulus and strength of lattice structures to their relative density:(5)$$\frac{E^{*}}{E_{s}}{=C}_{1}{\left(\frac{\rho^{*}}{\rho_{s}}\right)}^{n_{1}}$$
(6)$$\frac{\sigma^{*}}{\sigma_{s}}{=C}_{2}{\left(\frac{\rho^{*}}{\rho_{s}}\right)}^{n_{2}}$$

C_1_, n_1_, C_2_ and n_2_ are constant values dependent on the unit cell topology and are experimentally derived.

The n exponent can be predicted on the basis of the lattice deformation behaviour. In stretching-dominated structures, both stiffness and strength scale linearly as a function of the relative density and are higher than that of bending-dominated structures in which the elastic modulus scales quadratically with the relative density, while the strength scales with a factor of 3/2 [3].

The results of the experimental tests can be summarized in a graph in which the relative elastic modulus and relative strength are plotted against relative density (Figure 17 and Figure 18). Through data interpolation, the constant values reported in Equations (5) and (6) can be calculated for a given lattice structure.

Many studies performed on Ti6Al4V lattice materials are consistent with predictions obtained by the Gibson–Ashby model. Gibson–Ashby parameters from compressive tests, performed on different topologies and reported in the literature, were collected and are presented in Table 4.

Both the elastic modulus and compressive strength increase with the relative density. Collected experimental data were plotted in the Gibson–Ashby diagram to compare mechanical properties as a function of the relative density. Figure 19 shows the elastic modulus of the lattice structures, since the material used in the collected papers is the Ti6Al4V alloy in all cases. Therefore, the elastic modulus of the lattice materials can be compared to that of human bone. Figure 20 shows the relative strength against the relative density.

## 6. Biomedical Device Case Studies

Ti6Al4V porous device performances are widely examined via in silico, in vitro and in vivo experiments. Ti6Al4V scaffolds are applied for the treatment of bone disorders in the mandible, shoulder, spine, hip and femur.

Gao et al. [129] investigated and optimized, by employing finite element analysis (FEA) and the bone “Mechnostat” theory, porous scaffolds for mandibular defects. They considered both the biomechanical behaviour and mechanobiological property of scaffolds and analysed four lattice configurations with three strut diameters. The results showed a strong correlation between lattice topology and load transmission capacity, while mechanical failure strongly depends on the strut size and configuration; moreover, the computational model results indicated that the optimized scaffold can provide an excellent mechanical environment for bone regeneration. Liu et al. [130] compared conventional prostheses with homogeneous structures to stress-optimized functionally graded devices. They investigated, in silico, the damage resistance of four scaffolds and proposed a novel gradient algorithm for lightweight mandibular devices. The results illustrated that the optimized device reduced the maximum stress by 24.48% and increased the porosity by 6.82%, providing a better solution for mandibular reconstruction.

Bittredge et al. [131] investigated the stress shielding of total shoulder implants. The purpose of the study was the design and optimization of a lattice-based implant to manage the stiffness of a humeral implant stem used in shoulder implant applications. The study applied a topology lattice-optimization tool to develop various cellular designs that filled the original design of a shoulder implant and were further analysed by means of FEA and experimental tests. The results indicated that the proposed cellular implant can be effectively applied as a total shoulder replacement.

Fogel et al. [132] applied in vitro testing to elucidate the relative contribution of a porous design to intervertebral device stiffness and subsidence performance. Four groups of titanium cages were created with a combination of a porous endplate and/or an internal lattice architecture. The cage stiffness was scaled down by 16.7% by the internal lattice architecture and by 16.6% by the porous endplates. The cage with both porous parts exhibited the lowest stiffness with a value of 40.4 kN/mm and a motion segment stiffness of 1976.8 N/mm for subsidence. The internal lattice architecture showed no significant impact on the motion segment stiffness, while the porous endplates significantly decreased this value. Several works evaluated, using computed tomography (CT), the fusion rate of Ti6Al4V cages and PEEK cages by comparing groups of patients who had undergone cage implantation. Cuzzocrea et al. [133] found a better fusion rate and prevalence of fusion in the group treated with Ti6Al4V cages. Nemoto et al. [134] found a 96% fusion rate in the Ti6Al4V group and 64% in the PEEK group after 12 months. At 24 months, the fusion rate in the Ti6Al4V group was increased to 100%, while that in the PEEK group showed a 76% fusion rate. They also observed vertebral osteolysis in 60% of the cases with non-union in the PEEK group. This unusual finding was not observed in the Ti6Al4V group. Tanida et al. [135] observed a postoperative bone union rate of 75.2% and 74.5% at 1 year and 82.8% and 80.4% at 2 years for Ti6Al4V and PEEK groups, respectively, concluding that the bone union rate did not significantly differ between the two groups. They reported that the formation of vertebral endplate cysts is helpful for the non-union prediction, confirming after CT scan observations, the usefulness of this parameter.

Abate et al. [56] focused on the design of an acetabular cup using vintile lattice material with different porosities and pore sizes. The acetabular cup was then optimized by adjusting the porosity to improve mechanical performance and reduce stress shielding. In silico and in vitro experiments were carried out, and results showed that the optimized implant presents a weight reduction of 69%, reduced the stress shielding, has a more uniform stress distribution and has an elastic modulus in the range of that of human bones.

Gok [136] developed a multi-lattice design by dividing the proximal zone of a hip implant stem into three parts. Due to the multi-lattice design, a weight reduction of 25.89% was obtained and the maximum von Mises stresses in the stem were reduced from 289 to 189 MPa. They obtained stress shielding signals by determining the change in strain energy per unit bone mass caused by the presence of the femoral hip implant stem and its ratio to intact bone. In the case of multi-lattice design implants, there is a significant increase in stress-shielding signals from different zones of the femur.

## 7. Concluding Remarks

Progress in AM has considerably increased the production of complex structures unachievable with traditional fabrication techniques, such as lattice structures. This has led to research into these structures for different engineering applications. In the biomedical field, lattice materials made of Ti6Al4V alloy are widely used for the fabrication of scaffolds for bone substitutes. This review collected the most significant findings of the last ten years, which are summarized as follows:Data analysis of the results of the pore size effect on the bone ingrowth of eleven studies showed a wide range of optimal pore sizes from 100 up to 1000 μm, with an optimal mean value of 522 μm. The comparison showed little discrepancies, since works that evaluated comparable ranges of pore sizes found different optimal results.The analysis of the effect of surface roughness showed that minimal differences in the roughness values do not affect the cell adhesion and proliferation. In other studies, the comparison of a wide range values from 3 to 8 μm showed that the optimal surface roughness values are between 6 and 8 μm.Findings on the effect of the elastic modulus showed that reducing the implant stiffness to that of human bone improves stress stimulation and reduces stress shielding. Several studies with implants having an elastic modulus comparable to that of human bone revealed no significant influence of elastic material properties on bone ingrowth.The Gibson–Ashby model is useful for comparing the mechanical performance of lattice structures and confirmed the suitability of the Ti6Al4V alloy for biomedical applications. Indeed, the collected results showed that the elastic modulus of the selected lattice materials, with relative densities under 30%, falls within the range of the cancellous bone elastic modulus.

Future works include the development of a repeatable and robust design methodology of biomedical implants with a combination of Gibson–Ashby model data and in silico and in vitro experiments. Data reported in the present review provide a scientific base for the choice of the optimal lattice topology and design parameters.

## Figures and Tables

**Figure 1 jfb-14-00125-f001:**
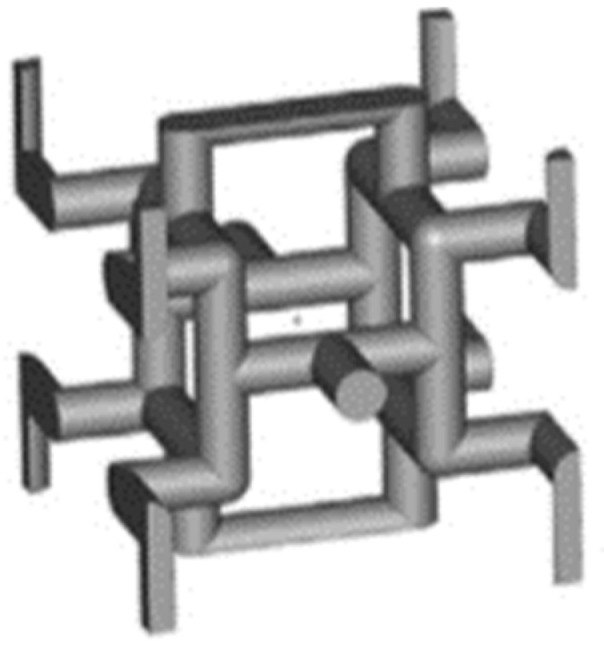
Gibson–Ashby lattice structure [3].

**Figure 2 jfb-14-00125-f002:**
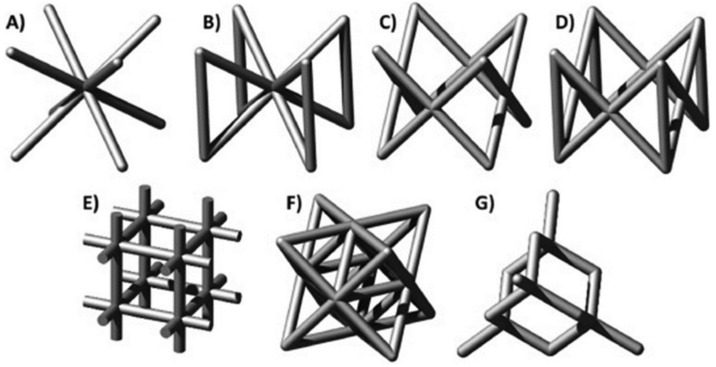
Strut-based lattice topologies: (**A**) BCC; (**B**) BCCZ; (**C**) FCC; (**D**) FCCZ; (**E**) cubic; (**F**) octet-truss; (**G**) diamond [29].

**Figure 3 jfb-14-00125-f003:**
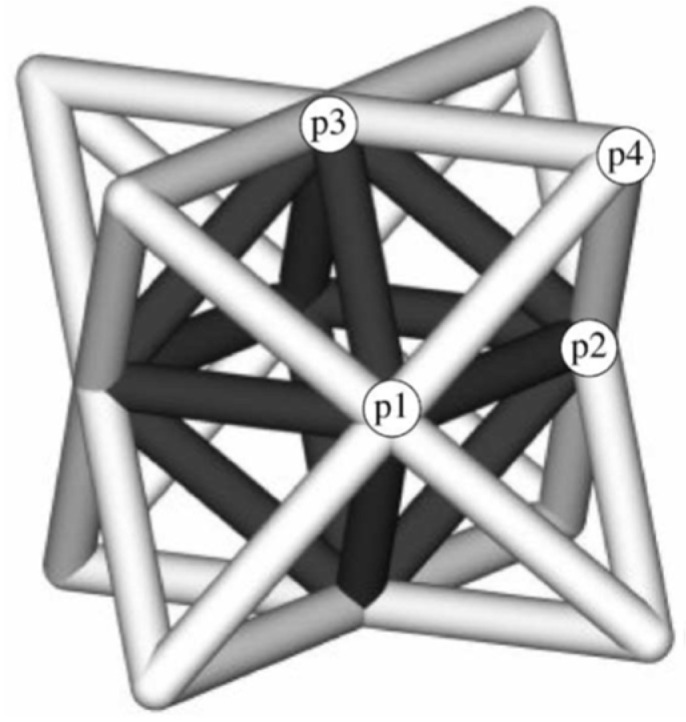
Structure of the octet-truss unit cell [30].

**Figure 4 jfb-14-00125-f004:**
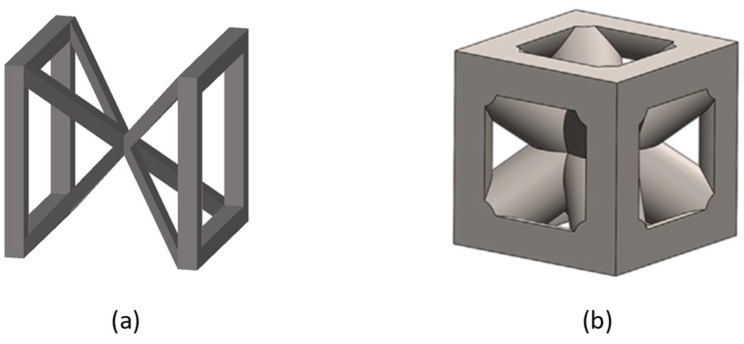
BCC-derived lattice structures: (**a**) G7; (**b**) SCBCC [31].

**Figure 5 jfb-14-00125-f005:**
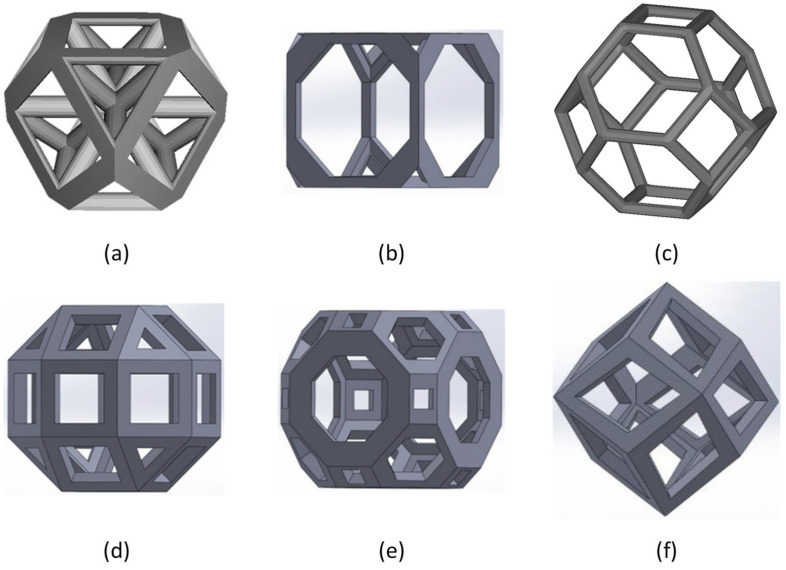
Archimedean and Catalan solid lattice structures: (**a**) cuboctahedron; (**b**) truncated cube; (**c**) truncated octahedron (Kelvin); (**d**) rhombicuboctahedron; (**e**) truncated cuboctahedron; (**f**) rhombic dodecahedron [33,35].

**Figure 6 jfb-14-00125-f006:**
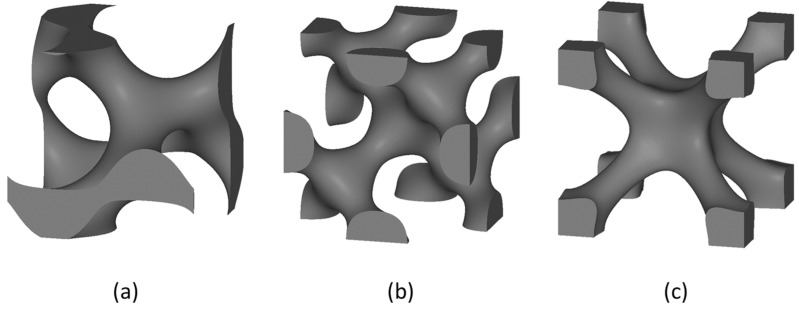
TPMS lattice structures: (**a**) skeletal gyroid; (**b**) skeletal diamond; (**c**) skeletal IWP; (**d**) sheet gyroid; (**e**) sheet diamond; (**f**) sheet IWP; (**g**) Schwarz primitive; (**h**) FRD; (**i**) Neovius.

**Figure 7 jfb-14-00125-f007:**
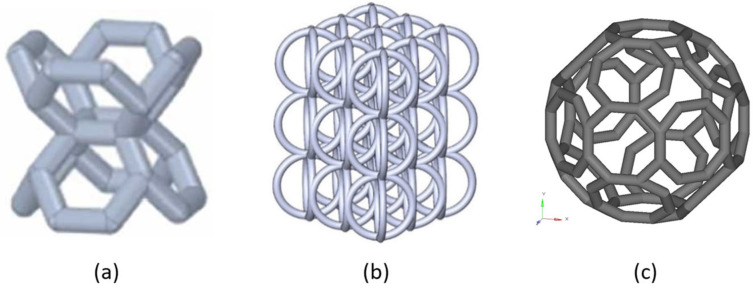
Novel designed lattice materials: (**a**) vintile cell; (**b**) circular cell-based lattice; (**c**) TAOR cell [57,58,59].

**Figure 8 jfb-14-00125-f008:**
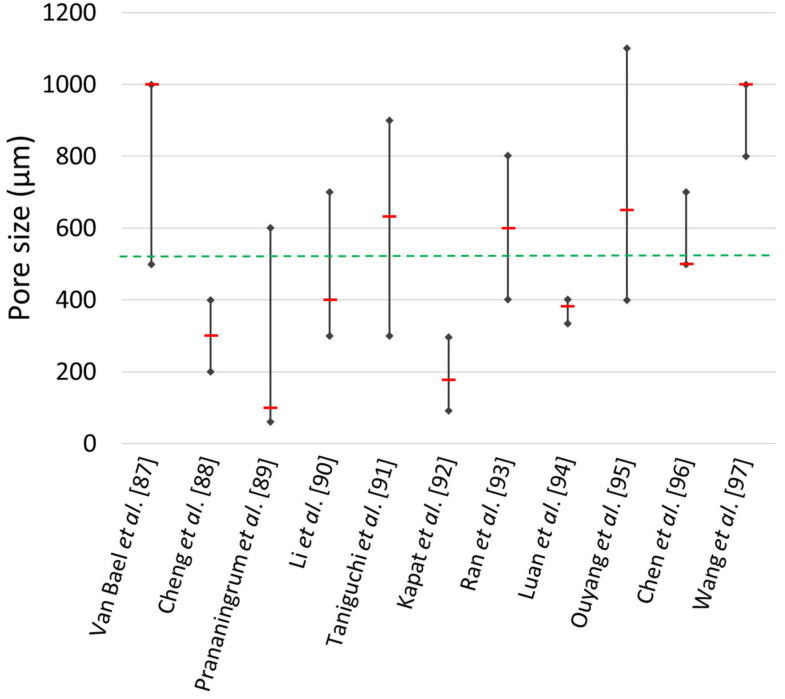
Evaluated optimal pore size for bone ingrowth.

**Figure 9 jfb-14-00125-f009:**
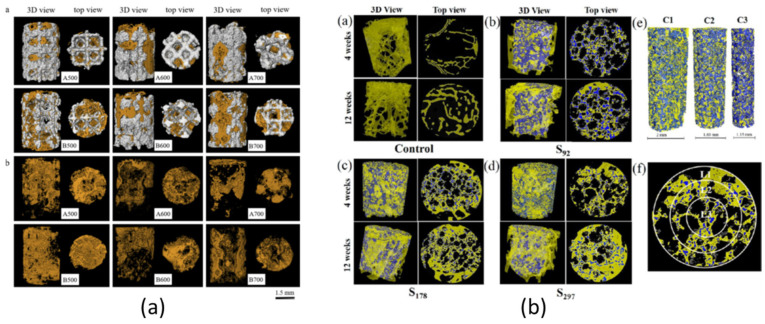
Quantitative μ-CT analysis of new bone ingrowth in the research papers of the following: (**a**) Chen et al. [96]; (**b**) Kapat et al. [92]; (**c**) Ouyang et al. [95].

**Figure 10 jfb-14-00125-f010:**
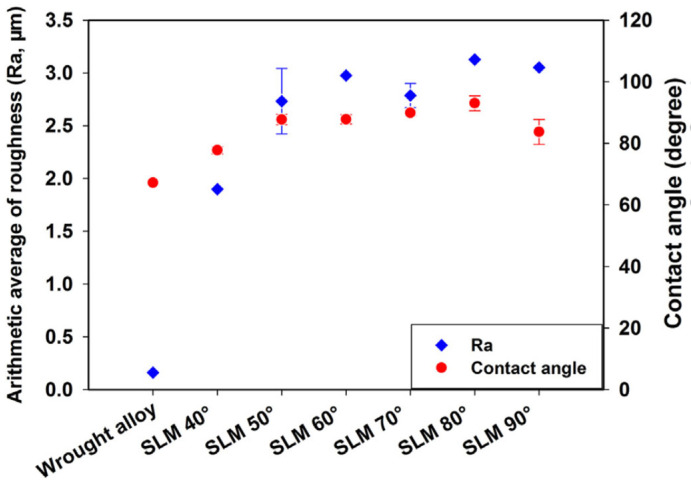
Arithmetic average roughness (Ra) and contact angle of wrought alloy and SLM-fabricated Ti6Al4V discs with different additive angles [100].

**Figure 11 jfb-14-00125-f011:**
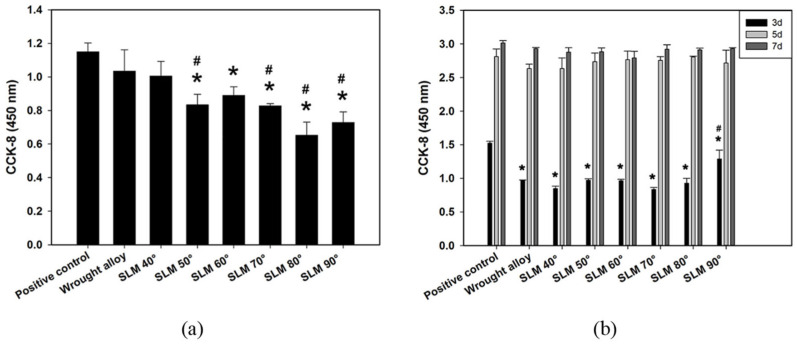
Cell (**a**) adhesion and (**b**) proliferation on wrought alloy and SLM-fabricated Ti6Al4V discs with different additive angles and surface roughness [100]. * indicates *p* < 0.5; # indicates *p* < 0.01.

**Figure 12 jfb-14-00125-f012:**
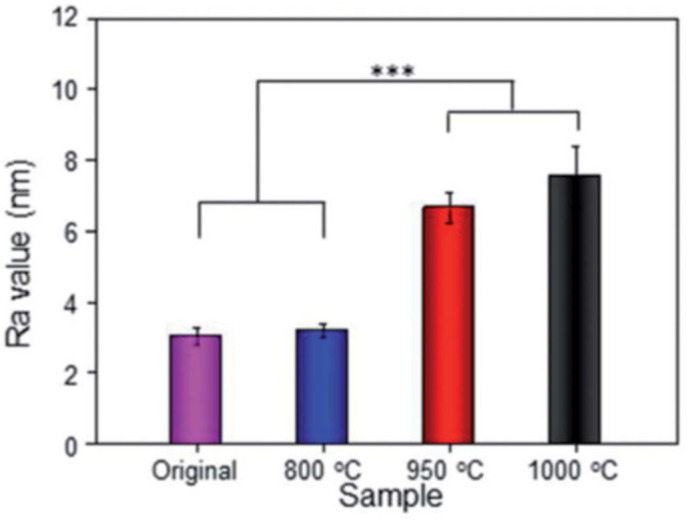
Ra values of original and heat-treated samples [101]. *** indicates *p* < 0.5.

**Figure 13 jfb-14-00125-f013:**
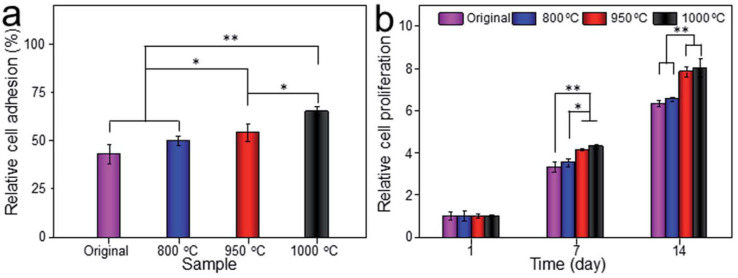
(**a**) Relative cell adhesion on the porous titanium cultured in vitro for 24 h; (**b**) relative cell proliferation on the porous titanium cultured in vitro for 1, 7 and 14 days [101]. * indicates *p* < 0.5; ** indicates *p* < 0.01.

**Figure 14 jfb-14-00125-f014:**
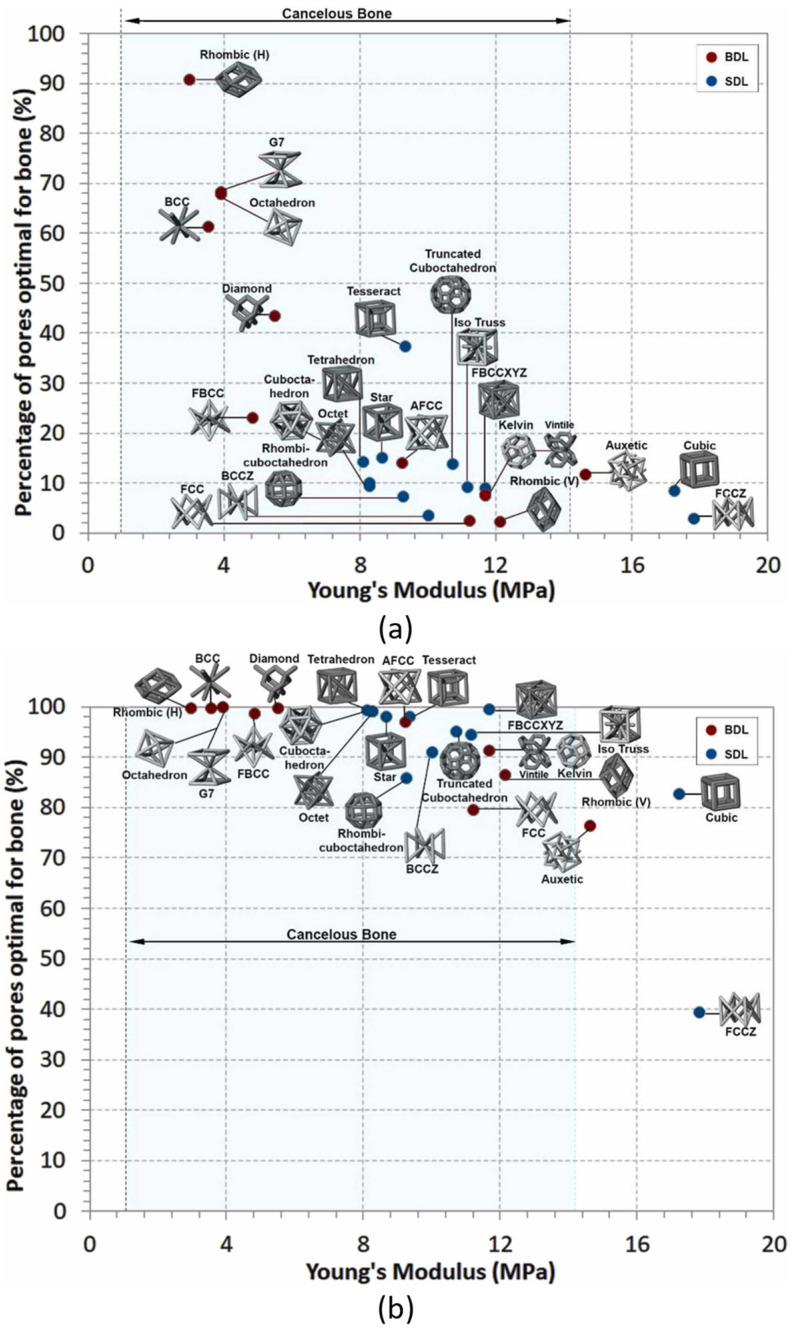
Percentage of pores optimal for bone ingrowth as a function of the elastic modulus under a pressure and a relative density of, respectively: (**a**) 0.5 MPa and 30%; (**b**) 2 MPa and 30% [59].

**Figure 15 jfb-14-00125-f015:**
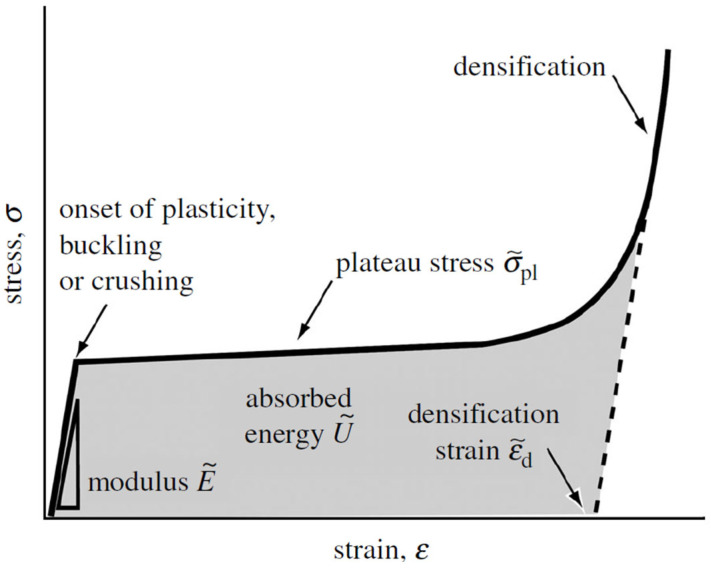
Stress–strain curve of a bending-dominated lattice [117].

**Figure 16 jfb-14-00125-f016:**
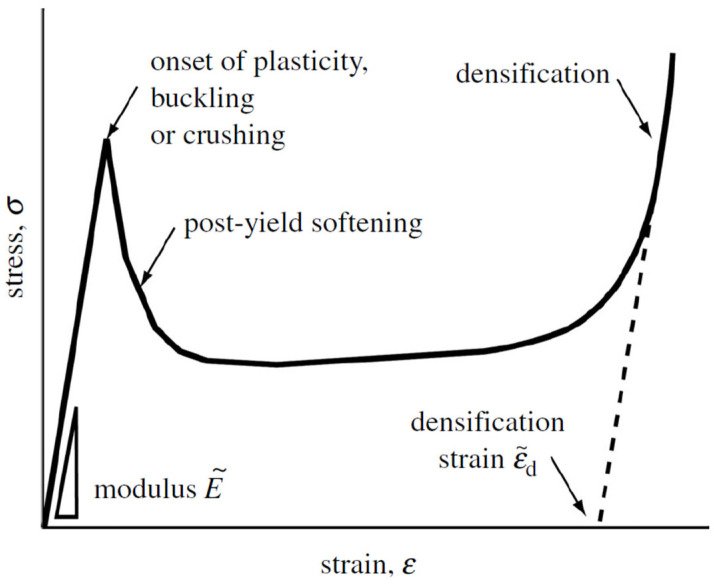
Stress–strain curve of a stretch-dominated lattice [117].

**Figure 17 jfb-14-00125-f017:**
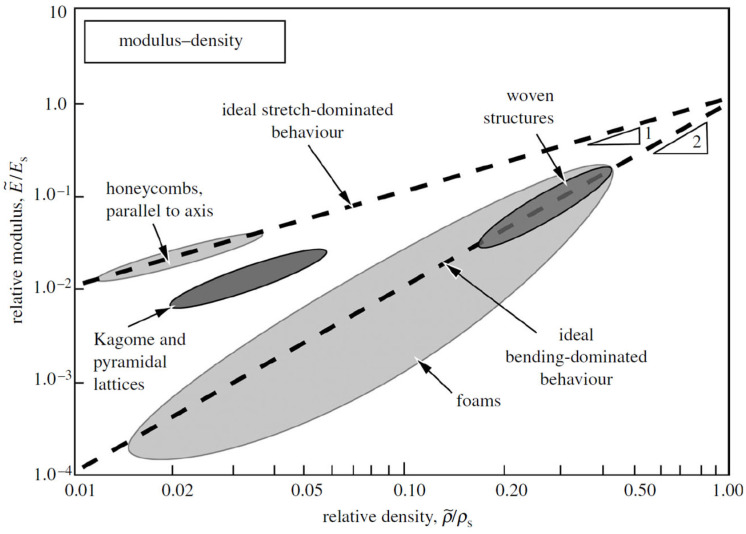
Gibson–Ashby model: relative modulus against relative density [117].

**Figure 18 jfb-14-00125-f018:**
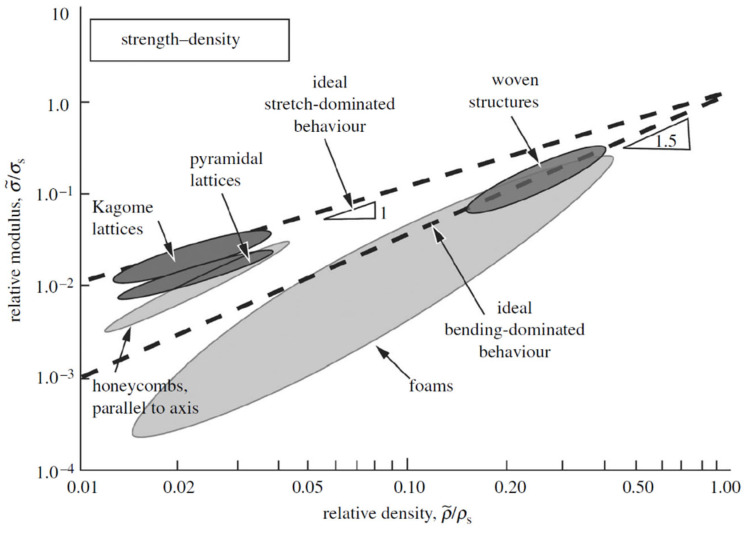
Gibson–Ashby model: relative strength against relative density [117].

**Figure 19 jfb-14-00125-f019:**
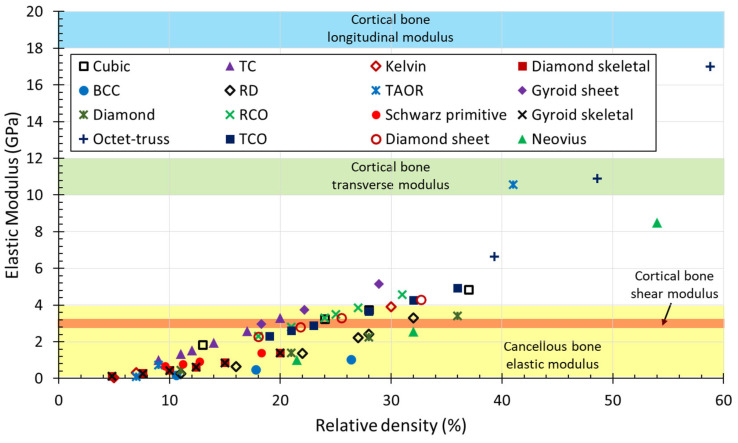
Comparison of the collected experimental data in the Gibson–Ashby diagram: elastic modulus against relative density.

**Figure 20 jfb-14-00125-f020:**
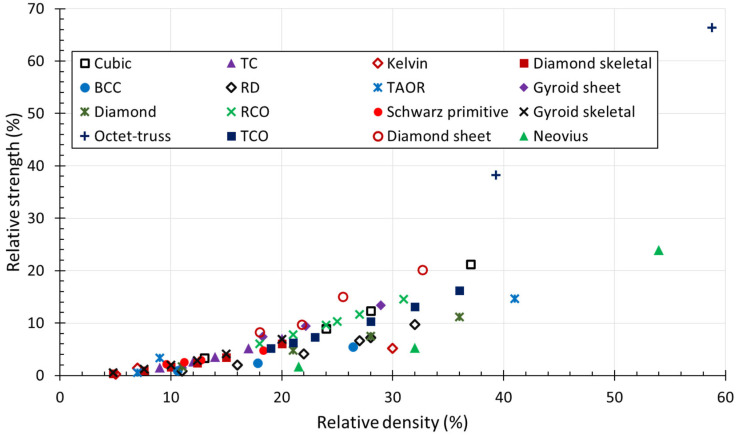
Comparison of the collected experimental data in the Gibson–Ashby diagram: relative strength against relative density.

**Table 1 jfb-14-00125-t001:** Examples of the most common TPMS equations.

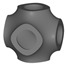	φprimitive≡cosx+cosy+cosz=C
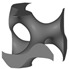	φgyroid≡sinxcosy+sinzcosx+sinycosz=C
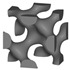	φdiamond≡cosxcosycosz-sinxsinysinz=C
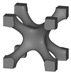	φIWP≡2(cosxcosy+cosycosz+coszcosx)-(cos2x+cos2y+cos2z)=C
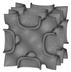	φFRD≡4(cosxcosycosz)-(cos2xcos2y+cos2ycos2z+cos2zcos2x)=C
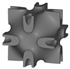	φNeovius≡3(cosx+cosy+cosz)+4cosxcosycosz=C

**Table 2 jfb-14-00125-t002:** Comparison of different research findings for the evaluation of the optimal pore size for bone ingrowth.

Research	Optimal Pore Size [μm]	Tested Pore Size Range [μm]
Van Bael et al. [87]	1000	500–1000
Cheng et al. [88]	300	200–400
Prananingrum et al. [89]	100	60–600
Li et al. [90]	400	300–700
Taniguchi et al. [91]	632	300–900
Kapat et al. [92]	178	92–297
Ran et al. [93]	600	400–800
Luan et al. [94]	383	334–401
Ouyang et al. [95]	650	400–1100
Chen et al. [96]	500	500–700
Wang et al. [97]	1000	800–1000

**Table 3 jfb-14-00125-t003:** Maxwell number for strut-based unit cells.

Unit Cell Topology	Cubic	BCC	Diamond	Octet-Truss	TC	RD	RCO	TCO
S	12	8	9	36	36	24	48	72
N	8	9	7	14	24	14	24	48
M	−6	−13	−6	0	−30	−12	−18	−66

**Table 4 jfb-14-00125-t004:** Gibson–Ashby model parameters from compressive tests reported in the literature.

Unit Cell Topology	Elastic Modulus (GPa)		Compressive Strength (MPa)	
	C_1_	n_1_	C_2_	n_2_
Cubic [35]	0.11	0.92	1.15	1.75
Cubic [119]	0.55	2.82	1.34	1.85
BCC [120]	0.15	2	0.57	1.9
BCC [121]	0.15	2	0.23	1.5
Diamond [118]	0.17	1.68	0.56	1.58
Octet-truss [119]	0.51	2.33	1.37	1.37
TC [118]	0.32	1.5	1.49	1.9
RD [118]	0.42	2.34	1.29	2.27
RD [122]	1.08	1.9	0.6	1.31
RCO [118]	0.17	1.25	0.97	1.62
TCO [118]	0.14	1.18	0.99	1.78
Kelvin [58]	0.6	2.3	0.3	1.5
TAOR [58]	0.8	2.3	0.6	1.5
Schwarz Primitive [123]	0.09	1.15	0.34	1.25
Schwarz Primitive [124]	1.38	2	0.98	1.5
Diamond Sheet [125]	0.71	1.21	0.42	1.14
Diamond Sheet [123]	0.12	1.06	1.66	1.89
Diamond Skeletal [15]	0.17	1.64	1.39	1.95
Diamond Skeletal [126]	0.7	2.7	1.17	2.6
Gyroid Sheet [127]	0.2	1.2	0.67	1.3
Gyroid Sheet [123]	0.12	1.1	2.07	2.03
Gyroid Skeletal [15]	0.19	1.71	1.31	1.83
Gyroid Skeletal [128]	0.29	2	0.46	1.5
Neovius [126]	0.31	2.3	1.43	2.9

## Data Availability

Not available.

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
