# Peer review of "Titanium Lattice Structures Produced via Additive Manufacturing for a Bone Scaffold: A Review"

_jfb, 2023, doi:10.3390/jfb14030125_

Round 1

Reviewer 1 Report

The aim of the paper is to provide a comprehensive view of the mechanical and morphological requirements of lattice structures for the design of biomedical implants for bone substitutes. The review is focused on the effect of pore size and surface roughness on the bone ingrowth, and on the effect of elastic modulus in the reduction of stress shielding and pro-61 motion of osseointegration. The topic is up-to-date and corresponds to the journal’s area. The abstract is well structured and informative enough. The structure of the manuscript is well designed. Adequate conclusions are drown.

There are a few remarks which should be edited:

1.     There are many abbreviations and it will be better if they are given in a table at the beginning of the manuscript.

2.     There are several notices: “Error! Reference not found”. Please find, check and edit.

Author Response

Reviewer #1 Report Form

Comments and Suggestions for Authors

The aim of the paper is to provide a comprehensive view of the mechanical and morphological requirements of lattice structures for the design of biomedical implants for bone substitutes. The review is focused on the effect of pore size and surface roughness on the bone ingrowth, and on the effect of elastic modulus in the reduction of stress shielding and pro-61 motion of osseointegration. The topic is up-to-date and corresponds to the journal’s area. The abstract is well structured and informative enough. The structure of the manuscript is well designed. Adequate conclusions are drown.

There are a few remarks which should be edited:

  1. There are many abbreviations and it will be better if they are given in a table at the beginning of the manuscript.

An abbreviation list have been added in the text (please, see lines from 24 to 60)

Nomenclature

TPMS   triply periodic minimal surface

AM       additive manufacturing

SLM     selective laser melting

DMLS   direct metal laser sintering

EBM     electron beam melting

BCC     body centred cubic

FCC     face centred cubic

BCCZ   body centred cubic with vertical struts

FCCZ   face centred cubic with vertical struts

SCBCC simple cubic body centred cubic

TC        truncated cube

TO       truncated octahedron

RCO     rhombicuboctahedron

TCO     truncated cuboctahedron

RD       rhombic dodecahedron

TAOR  triply arranged octagonal rings

τ           complex variable

θ          Bonnet angle

R(τ)      TPMS function

γ          position vector in the Euclidean space

Ak         amplitude factor

hk          kth grid vector in the reciprocal space

λ        periodic wavelength

pk         phase offset

C          constant factor related to porosity

M         Maxwell number

S          number of struts

N          number of nodes

E*         elastic modulus of the lattice structure

σ*         compressive strength of the lattice structure

Es         elastic modulus of the parent material

σs         compressive strength of the parent material

FEA     finite element analysis

PEEK    polyetheretherketone

CT        computed tomography

  1. There are several notices: “Error! Reference not found”. Please find, check and edit.

Authors are sorry for the error, all the references have been fixed

Reviewer 2 Report

The study is incomplete and lacks significant studies done in this field.

The lattice structure size and geometry is crucial for angiogenesis. Nothing has been discussed in such a regime.

It is not a comprehensive study on the topic, and in such respect should only discuss the current status of lattice geometry in titanium implants biomedical engineering. 

The study should discuss the importance of lattice geometry in terms with different applications - such as hip, knee, spine implants. 

The study should also discuss the manufacturing regime of such lattice structures, and discuss the difficulty in generation of complex geometries with adequate reproducibility.

Author Response

Reviewer#2 Report Form

Comments and Suggestions for Authors

The study is incomplete and lacks significant studies done in this field.

Novel studies have been added in the text:

  • Caiazzo, F.; Alfieri, V.; Brahim, &; Bujazha, D. Additive Manufacturing of Biomorphic Scaffolds for Bone Tissue Engineering. The International Journal of Advanced Manufacturing Technology 2021, 113, 2909–2923, doi:10.1007/s00170-021-06773-5/Published.
  • Dong, G.; Zhao, Y.F. Numerical and Experimental Investigation of the Joint Stiffness in Lattice Structures Fabricated by Additive Manufacturing. Int J Mech Sci 2018, 148, 475–485, doi:10.1016/j.ijmecsci.2018.09.014.
  • Abate, K.M.; Nazir, A.; Yeh, Y.P.; Chen, J.E.; Jeng, J.Y. Design, Optimization, and Validation of Mechanical Properties of Different Cellular Structures for Biomedical Application. International Journal of Advanced Manufacturing Technology 2020, 106, 1253–1265, doi:10.1007/s00170-019-04671-5.
  • Abate, K.M.; Nazir, A.; Chen, J.E.; Jeng, J.Y. Design, Optimization, and Evaluation of Additively Manufactured Vintiles Cellular Structure for Acetabular Cup Implant. Processes 2020, 8, doi:10.3390/pr8010025.
  • Alomar, Z.; Concli, F. Compressive Behavior Assessment of a Newly Developed Circular Cell-Based Lattice Structure. Mater Des 2021, 205, doi:10.1016/j.matdes.2021.109716.
  • Distefano, F.; Mineo, R.; Epasto, G. Mechanical Behaviour of a Novel Biomimetic Lattice Structure for Bone Scaffold. J Mech Behav Biomed Mater 2023, 138, doi:10.1016/j.jmbbm.2023.105656.
  • Parisien, A.; ElSayed, M.S.A.; Frei, H. Mechanoregulation Modelling of Stretching versus Bending Dominated Periodic Cellular Solids. Mater Today Commun 2022, 33, doi:10.1016/j.mtcomm.2022.104315.
  • Ghouse, S.; Babu, S.; van Arkel, R.J.; Nai, K.; Hooper, P.A.; Jeffers, J.R.T. The Influence of Laser Parameters and Scanning Strategies on the Mechanical Properties of a Stochastic Porous Material. Mater Des 2017, 131, 498–508, doi:10.1016/j.matdes.2017.06.041.
  • Gao, W.; Zhang, Y.; Ramanujan, D.; Ramani, K.; Chen, Y.; Williams, C.B.; Wang, C.C.L.; Shin, Y.C.; Zhang, S.; Zavattieri, P.D. The Status, Challenges, and Future of Additive Manufacturing in Engineering. CAD Computer Aided Design 2015, 69, 65–89, doi:10.1016/j.cad.2015.04.001.
  • Lin, K.; Yuan, L.; Gu, D. Influence of Laser Parameters and Complex Structural Features on the Bio-Inspired Complex Thin-Wall Structures Fabricated by Selective Laser Melting. J Mater Process Technol 2019, 267, 34–43, doi:10.1016/j.jmatprotec.2018.12.004.
  • Sachs E.M. U.S. Patent No. 6,036,777. Washington, DC: U.S. Patent and Trademark Office. 2000.
  • Zhang, L.; Zhang, S.; Zhu, H.; Hu, Z.; Wang, G.; Zeng, X. Horizontal Dimensional Accuracy Prediction of Selective Laser Melting. Mater Des 2018, 160, 9–20, doi:10.1016/j.matdes.2018.08.059.
  • Nawada, S.; Dimartino, S.; Fee, C. Dispersion Behavior of 3D-Printed Columns with Homogeneous Microstructures Comprising Differing Element Shapes. Chem Eng Sci 2017, 164, 90–98, doi:10.1016/j.ces.2017.02.012.
  • Abdulhameed, O.; Al-Ahmari, A.; Ameen, W.; Mian, S.H. Additive Manufacturing: Challenges, Trends, and Applications. Advances in Mechanical Engineering 2019, 11, doi:10.1177/1687814018822880.
  • Abdulmaged, A.I.; Soon, C.F.; Talip, B.A.; Zamhuri, S.A.A.; Mostafa, S.A.; Zhou, W. Characterization of Alginate–Gelatin–Cholesteryl Ester Liquid Crystals Bioinks for Extrusion Bioprinting of Tissue Engineering Scaffolds. Polymers (Basel) 2022, 14, doi:10.3390/polym14051021.
  • Cucinotta, F.; Mineo, R.; Raffaele, M.; Salmeri, F.; Sfravara, F. Customized Implant of Cervical Prostheses Exploiting a Predictive Analysis of Range of Motion. Comput Aided Des Appl 2023, 20, 122–133, doi:10.14733/cadaps.2023.S6.122-133.
  • Distefano, F.; Epasto, G.; Guglielmino, E.; Amata, A.; Mineo, R. Subsidence of a Partially Porous Titanium Lumbar Cage Produced by Electron Beam Melting Technology. J Biomed Mater Res B Appl Biomater 2023, 111, 590–598, doi:10.1002/jbm.b.35176.
  • Che, Z.; Sun, Y.; Luo, W.; Zhu, L.; Li, Y.; Zhu, C.; Liu, T.; Huang, L. Bifunctionalized Hydrogels Promote Angiogenesis and Osseointegration at the Interface of Three-Dimensionally Printed Porous Titanium Scaffolds. Mater Des 2022, 223, doi:10.1016/j.matdes.2022.111118.
  • Zhao, H.; Shen, S.; Zhao, L.; Xu, Y.; Li, Y.; Zhuo, N. 3D Printing of Dual-Cell Delivery Titanium Alloy Scaffolds for Improving Osseointegration through Enhancing Angiogenesis and Osteogenesis. BMC Musculoskelet Disord 2021, 22, doi:10.1186/s12891-021-04617-7.
  • Gao, P.; Fan, B.; Yu, X.; Liu, W.; Wu, J.; Shi, L.; Yang, D.; Tan, L.; Wan, P.; Hao, Y.; et al. Biofunctional Magnesium Coated Ti6Al4V Scaffold Enhances Osteogenesis and Angiogenesis in Vitro and in Vivo for Orthopedic Application. Bioact Mater 2020, 5, 680–693, doi:10.1016/j.bioactmat.2020.04.019.
  • Xu, X.; Lu, Y.; Li, S.; Guo, S.; He, M.; Luo, K.; Lin, J. Copper-Modified Ti6Al4V Alloy Fabricated by Selective Laser Melting with pro-Angiogenic and Anti-Inflammatory Properties for Potential Guided Bone Regeneration Applications. Materials Science and Engineering C 2018, 90, 198–210, doi:10.1016/j.msec.2018.04.046.
  • Lv, J.; Xiu, P.; Tan, J.; Jia, Z.; Cai, H.; Liu, Z. Enhanced Angiogenesis and Osteogenesis in Critical Bone Defects by the Controlled Release of BMP-2 and VEGF: Implantation of Electron Beam Melting-Fabricated Porous Ti6Al4V Scaffolds Incorporating Growth Factor-Doped Fibrin Glue. Biomedical Materials 2015, 10, doi:10.1088/1748-6041/10/3/035013
  • Bittredge, O.; Hassanin, H.; El-Sayed, M.A.; Eldessouky, H.M.; Alsaleh, N.A.; Alrasheedi, N.H.; Essa, K.; Ahmadein, M. Fabrication and Optimisation of Ti-6Al-4V Lattice-Structured Total Shoulder Implants Using Laser Additive Manufacturing. Materials 2022, 15, doi:10.3390/ma15093095.
  • Fogel, G.; Martin, N.; Lynch, K.; Pelletier, M.H.; Wills, D.; Wang, T.; Walsh, W.R.; Williams, G.M.; Malik, J.; Peng, Y.; et al. Subsidence and Fusion Performance of a 3D-Printed Porous Interbody Cage with Stress-Optimized Body Lattice and Microporous Endplates - a Comprehensive Mechanical and Biological Analysis. Spine Journal 2022, 22, 1028–1037, doi:10.1016/j.spinee.2022.01.003.
  • Cuzzocrea, F.; Ivone, A.; Jannelli, E.; Fioruzzi, A.; Ferranti, E.; Vanelli, R.; Benazzo, F. PEEK versus Metal Cages in Posterior Lumbar Interbody Fusion: A Clinical and Radiological Comparative Study. Musculoskelet Surg 2019, 103, 237–241, doi:10.1007/s12306-018-0580-6.
  • Nemoto, O.; Asazuma, T.; Yato, Y.; Imabayashi, H.; Yasuoka, H.; Fujikawa, A. Comparison of Fusion Rates Following Transforaminal Lumbar Interbody Fusion Using Polyetheretherketone Cages or Titanium Cages with Transpedicular Instrumentation. European Spine Journal 2014, 23, 2150–2155, doi:10.1007/s00586-014-3466-9.
  • Tanida, S.; Fujibayashi, S.; Otsuki, B.; Masamoto, K.; Takahashi, Y.; Nakayama, T.; Matsuda, S. Vertebral Endplate Cyst as a Predictor of Nonunion after Lumbar Interbody Fusion: Comparison of Titanium And Polyetheretherketone Cages. Spine (Phila Pa 1976) 2016, 152, 28, doi:10.1097/BRS.0000000000001605.
  • Gok, M.G. Creation and Finite-Element Analysis of Multi-Lattice Structure Design in Hip Stem Implant to Reduce the Stress-Shielding Effect. Proceedings of the Institution of Mechanical Engineers, Part L: Journal of Materials: Design and Applications 2022, 236, 429–439, doi:10.1177/14644207211046200.

The lattice structure size and geometry is crucial for angiogenesis. Nothing has been discussed in such a regime.

Angiogenesis have been discussed in the text (please see lines from 219 to 227)

In addition, angiogenesis is an essential physiological process for bone regeneration [79]; the biological inertia of Ti6Al4V surface and the deficit of angiogenesis, may cause postoperative complications such as dislocation or loosening of the device [80]. Several methods to enhance angiogenesis of Ti6Al4V scaffolds were evaluated in literature, including: the development of multifunctional surface coatings with angiogenic properties [81]; the incorporation into Ti6Al4V alloy of copper ions, which presents high bioactivity and outstanding antibacterial properties [82]; the controlled release of bone-morphogenic protein-2 and vascular endothelial growth factor in the Ti6Al4V alloy [83].

It is not a comprehensive study on the topic, and in such respect should only discuss the current status of lattice geometry in titanium implants biomedical engineering.

The authors tried to improve the manuscript by adding new sections about the following topics:

- applications of lattice structures in biomedical devices;

- limits of additive manufacturing technology for producing biomedical devices including lattice structures;

- angiogenesis was discussed by adding some papers on this topic.

Moreover, the authors added more relevant papers of the topic of the review.

The study should discuss the importance of lattice geometry in terms with different applications - such as hip, knee, spine implants.

A case study section has been added in the text (please see lines from 459 to 517)

  1. Biomedical devices case studies

Ti6Al4V porous devices performance are widely examined by in silico, in vitro and in vivo experiments. Ti6Al4V scaffolds are applied for the treatment of bone disorders in mandible, shoulder, spine, hip, and femur.     
Gao et al. [130] investigated and optimised, by employing finite element analysis (FEA) and bone “Mechnostat” theory, porous scaffolds for mandibular defects. They considered both the biomechanical behaviour and mechanobiological property of scaffolds and analysed four lattice configurations with three strut diameters. The results showed a strong correlation between lattice topology and load transmission capacity, while mechanical failure strongly depends on strut size and configuration; moreover, the computational model results indicated that the optimised scaffold can provide an excellent mechanical environment for bone regeneration. Liu et al. [131] compared conventional prostheses with homogeneous structures to stress-optimised functionally graded devices. They in silico investigated the damage resistance of four porous scaffolds and proposed a novel gradient algorithm for lightweight mandibular prostheses. The results illustrated that compared with the initial homogeneous prosthesis, the optimised device reduced the maximum stress by 24.48% and increased the porosity by 6.82%, providing a better solution for mandibular reconstruction.           
Bittredge et al. [132] investigated the stress shielding of total shoulder implants. The objective of the study was the design and optimization of a lattice-based implant to control the stiffness of a humeral implant stem used in shoulder implant applications. The study used a topology lattice-optimisation tool to create different cellular designs that filled the original design of a shoulder implant and were further analysed by means of FEA and experimental tests. The results indicated that the proposed cellular implant can be effectively applied as total shoulder replacement.        
Fogel et al. [133] applied in vitro testing to elucidate the relative contribution of a porous design to intervertebral device stiffness and subsidence performance. Four groups of titanium cages were created with a combination of a porous endplate and/or an internal lattice architecture. The cage stiffness was reduced by 16.7% by the internal lattice architecture, and by 16.6% by the porous endplates. The cage with both porous parts showed the lowest stiffness with a value of 40.4 kN/mm and a motion segment stiffness of 1976.8 N/mm for subsidence. The internal lattice architecture showed no significant impact on the motion segment stiffness while the porous endplates significantly decreased this value. Several works evaluated, using Computed Tomography (CT), the fusion rate of Ti6Al4V cages and PEEK cages by comparing groups of patients who have undergone cage implantation. Cuzzocrea et al. [134] found a better fusion rate and prevalence of fusion in the group treated with Ti6Al4V cages. Nemoto et al. [135] found 96% of fusion rate in the Ti6Al4V group and 64 % in the PEEK group after 12 months. At 24 months, fusion rate in the Ti6Al4V group was increased to 100%, while PEEK group showed 76% of fusion rate. They also observed vertebral osteolysis in the 60% of the cases with nonunion in the PEEK group. This abnormal finding was not observed in the Ti6Al4V group. Tanida et al. [136] observed postoperative bone union rate of 75.2% and 74.5% at 1 year, and 82.8% and 80.4% at 2 years for Ti6Al4V and PEEK groups respectively, concluding that bone union rate did not significantly differ between the two groups. They reported that the formation of vertebral endplate cysts is useful for the nonunion prediction, confirming after CT scans observation the usefulness of this parameter.    
Abate et al. [55] focused on the design of an acetabular cup using vintile lattice material with different porosities and pore sizes. The acetabular cup was then optimised by adjusting the porosity to improve mechanical performance and reduce stress shielding. In silico and in vitro experiments were carried out and results showed that optimised implant presents a weight reduction of 69%, reduced the stress shielding, has a more uniform stress distribution and has an elastic modulus in the range of the human bones.         
Gok [137] developed a multi-lattice design by dividing into three parts the proximal zone of a hip implant stem. Due to the multi-lattice design, a weight reduction of 25.89% was obtained and the maximum von Mises stresses in the stem were reduced from 289 to 189 MPa. Author obtained stress shielding signals by determining the change in strain energy per unit bone mass caused by the presence of the femoral hip implant stem and its ratio to intact bone. In the case of multi-lattice design implants, there is a significant increase in stress-shielding signals from different zones of the femur.

The study should also discuss the manufacturing regime of such lattice structures, and discuss the difficulty in generation of complex geometries with adequate reproducibility.

A section on the current status of additive manufacturing technologies has been added in the text (please see lines from 175 to 205)

  1. Current status of Additive Manufacturing technologies

Additive Manufacturing (AM) has grown considerably in recent years, with improvements in technology and resulting material properties. The ability to create components with complex parts with customisable material properties is one of the most important advantages of this technology, allowing the production of complex functional objects from various materials unattainable by conventional manufacturing methods [5]. This has led to the industrial use of AM parts even in highly critical applications, most notably in aerospace, automotive and biomedical applications. Different AM technologies are currently used for the fabrication of parts used in this fields, from metallic fine powders; for instance: Selective Laser Melting (SLM) [7], Direct Metal Laser Sintering (DMLS) [8], Electron Beam Melting (EBM) [9].    
However, due to the rapid proliferation of a wide variety of technologies associated with AM, there is a lack of a comprehensive set of design principles, manufacturing guidelines, and standardization of best practices. AM techniques require process optimization and quality control to ensure accuracy and reliability [59]. This requirement is critically important for parts with complex geometries, such as lattice structures, which include curved surfaces and thin connecting features. Different factors such as machine selection, processes and materials, orientation and position of the geometry, and finishing can alter the resulting quality of the printed part [60]. A major limitation is the minimum feature size for the AM system used [61], the achievable feature resolution is inherently constrained by the fact that powder-based systems require particles that are larger than 20 μm so that the powder can be successfully spread during the recoating step [62]. An additional limitation is placed on the part design, most notably the build angles [63]. When extremely complex structure such as truncated icosahedra are printed with dimensions in the order of micrometres, some feature cannot be reproduced [64]. An important attribute is the surface quality, which is mainly determined by the thickness of each printed layer. Surface quality is also dependent on the form of the raw material; powder bed AM processes have poorer surface quality than others due to large and partially melted powder particles that reside on the printed part’s surface [60].        
In order to advance research interest and investments, AM technologies goal is to face these and other challenges to ensure the quality of the 3D printed products [65].

Reviewer 3 Report

The work is interesting, however, some revisions are needed before it can be accepted for publication:

There are some review papers focusing on the additive manufacturing of Titanium lattice structures for bone scaffold, please specify the novelty of this review paper.

The literature review should include some of the recent papers published in mdpi journals, such as 10.3390/polym14051021; 10.1007/s00170-021-06773-5

Judging from the current state of the art, what are the future development trends? What scientific methods will be more thoroughly studied in the future? These should be presented at the last.

Author Response

Reviewer#3 Report Form

Comments and Suggestions for Authors

The work is interesting, however, some revisions are needed before it can be accepted for publication:

There are some review papers focusing on the additive manufacturing of Titanium lattice structures for bone scaffold, please specify the novelty of this review paper.

The current work intends to furnish guidelines in the choice of the most suitable lattice topology by applying the Gibson-Ashby model and by focusing on the performance and features needed for the bone’s stimulation in the osseointegration process. (please, see lines from 98 to 101)

The literature review should include some of the recent papers published in mdpi journals, such as 10.3390/polym14051021; 10.1007/s00170-021-06773-5

Authors thanks the reviewer for the suggestion, the recommended papers have been added in the text. A full list of novel papers added in the text is presented as follows:

  • Caiazzo, F.; Alfieri, V.; Brahim, &; Bujazha, D. Additive Manufacturing of Biomorphic Scaffolds for Bone Tissue Engineering. The International Journal of Advanced Manufacturing Technology 2021, 113, 2909–2923, doi:10.1007/s00170-021-06773-5/Published.
  • Dong, G.; Zhao, Y.F. Numerical and Experimental Investigation of the Joint Stiffness in Lattice Structures Fabricated by Additive Manufacturing. Int J Mech Sci 2018, 148, 475–485, doi:10.1016/j.ijmecsci.2018.09.014.
  • Abate, K.M.; Nazir, A.; Yeh, Y.P.; Chen, J.E.; Jeng, J.Y. Design, Optimization, and Validation of Mechanical Properties of Different Cellular Structures for Biomedical Application. International Journal of Advanced Manufacturing Technology 2020, 106, 1253–1265, doi:10.1007/s00170-019-04671-5.
  • Abate, K.M.; Nazir, A.; Chen, J.E.; Jeng, J.Y. Design, Optimization, and Evaluation of Additively Manufactured Vintiles Cellular Structure for Acetabular Cup Implant. Processes 2020, 8, doi:10.3390/pr8010025.
  • Alomar, Z.; Concli, F. Compressive Behavior Assessment of a Newly Developed Circular Cell-Based Lattice Structure. Mater Des 2021, 205, doi:10.1016/j.matdes.2021.109716.
  • Distefano, F.; Mineo, R.; Epasto, G. Mechanical Behaviour of a Novel Biomimetic Lattice Structure for Bone Scaffold. J Mech Behav Biomed Mater 2023, 138, doi:10.1016/j.jmbbm.2023.105656.
  • Parisien, A.; ElSayed, M.S.A.; Frei, H. Mechanoregulation Modelling of Stretching versus Bending Dominated Periodic Cellular Solids. Mater Today Commun 2022, 33, doi:10.1016/j.mtcomm.2022.104315.
  • Ghouse, S.; Babu, S.; van Arkel, R.J.; Nai, K.; Hooper, P.A.; Jeffers, J.R.T. The Influence of Laser Parameters and Scanning Strategies on the Mechanical Properties of a Stochastic Porous Material. Mater Des 2017, 131, 498–508, doi:10.1016/j.matdes.2017.06.041.
  • Gao, W.; Zhang, Y.; Ramanujan, D.; Ramani, K.; Chen, Y.; Williams, C.B.; Wang, C.C.L.; Shin, Y.C.; Zhang, S.; Zavattieri, P.D. The Status, Challenges, and Future of Additive Manufacturing in Engineering. CAD Computer Aided Design 2015, 69, 65–89, doi:10.1016/j.cad.2015.04.001.
  • Lin, K.; Yuan, L.; Gu, D. Influence of Laser Parameters and Complex Structural Features on the Bio-Inspired Complex Thin-Wall Structures Fabricated by Selective Laser Melting. J Mater Process Technol 2019, 267, 34–43, doi:10.1016/j.jmatprotec.2018.12.004.
  • Sachs E.M. U.S. Patent No. 6,036,777. Washington, DC: U.S. Patent and Trademark Office. 2000.
  • Zhang, L.; Zhang, S.; Zhu, H.; Hu, Z.; Wang, G.; Zeng, X. Horizontal Dimensional Accuracy Prediction of Selective Laser Melting. Mater Des 2018, 160, 9–20, doi:10.1016/j.matdes.2018.08.059.
  • Nawada, S.; Dimartino, S.; Fee, C. Dispersion Behavior of 3D-Printed Columns with Homogeneous Microstructures Comprising Differing Element Shapes. Chem Eng Sci 2017, 164, 90–98, doi:10.1016/j.ces.2017.02.012.
  • Abdulhameed, O.; Al-Ahmari, A.; Ameen, W.; Mian, S.H. Additive Manufacturing: Challenges, Trends, and Applications. Advances in Mechanical Engineering 2019, 11, doi:10.1177/1687814018822880.
  • Abdulmaged, A.I.; Soon, C.F.; Talip, B.A.; Zamhuri, S.A.A.; Mostafa, S.A.; Zhou, W. Characterization of Alginate–Gelatin–Cholesteryl Ester Liquid Crystals Bioinks for Extrusion Bioprinting of Tissue Engineering Scaffolds. Polymers (Basel) 2022, 14, doi:10.3390/polym14051021.
  • Cucinotta, F.; Mineo, R.; Raffaele, M.; Salmeri, F.; Sfravara, F. Customized Implant of Cervical Prostheses Exploiting a Predictive Analysis of Range of Motion. Comput Aided Des Appl 2023, 20, 122–133, doi:10.14733/cadaps.2023.S6.122-133.
  • Distefano, F.; Epasto, G.; Guglielmino, E.; Amata, A.; Mineo, R. Subsidence of a Partially Porous Titanium Lumbar Cage Produced by Electron Beam Melting Technology. J Biomed Mater Res B Appl Biomater 2023, 111, 590–598, doi:10.1002/jbm.b.35176.
  • Che, Z.; Sun, Y.; Luo, W.; Zhu, L.; Li, Y.; Zhu, C.; Liu, T.; Huang, L. Bifunctionalized Hydrogels Promote Angiogenesis and Osseointegration at the Interface of Three-Dimensionally Printed Porous Titanium Scaffolds. Mater Des 2022, 223, doi:10.1016/j.matdes.2022.111118.
  • Zhao, H.; Shen, S.; Zhao, L.; Xu, Y.; Li, Y.; Zhuo, N. 3D Printing of Dual-Cell Delivery Titanium Alloy Scaffolds for Improving Osseointegration through Enhancing Angiogenesis and Osteogenesis. BMC Musculoskelet Disord 2021, 22, doi:10.1186/s12891-021-04617-7.
  • Gao, P.; Fan, B.; Yu, X.; Liu, W.; Wu, J.; Shi, L.; Yang, D.; Tan, L.; Wan, P.; Hao, Y.; et al. Biofunctional Magnesium Coated Ti6Al4V Scaffold Enhances Osteogenesis and Angiogenesis in Vitro and in Vivo for Orthopedic Application. Bioact Mater 2020, 5, 680–693, doi:10.1016/j.bioactmat.2020.04.019.
  • Xu, X.; Lu, Y.; Li, S.; Guo, S.; He, M.; Luo, K.; Lin, J. Copper-Modified Ti6Al4V Alloy Fabricated by Selective Laser Melting with pro-Angiogenic and Anti-Inflammatory Properties for Potential Guided Bone Regeneration Applications. Materials Science and Engineering C 2018, 90, 198–210, doi:10.1016/j.msec.2018.04.046.
  • Lv, J.; Xiu, P.; Tan, J.; Jia, Z.; Cai, H.; Liu, Z. Enhanced Angiogenesis and Osteogenesis in Critical Bone Defects by the Controlled Release of BMP-2 and VEGF: Implantation of Electron Beam Melting-Fabricated Porous Ti6Al4V Scaffolds Incorporating Growth Factor-Doped Fibrin Glue. Biomedical Materials 2015, 10, doi:10.1088/1748-6041/10/3/035013
  • Bittredge, O.; Hassanin, H.; El-Sayed, M.A.; Eldessouky, H.M.; Alsaleh, N.A.; Alrasheedi, N.H.; Essa, K.; Ahmadein, M. Fabrication and Optimisation of Ti-6Al-4V Lattice-Structured Total Shoulder Implants Using Laser Additive Manufacturing. Materials 2022, 15, doi:10.3390/ma15093095.
  • Fogel, G.; Martin, N.; Lynch, K.; Pelletier, M.H.; Wills, D.; Wang, T.; Walsh, W.R.; Williams, G.M.; Malik, J.; Peng, Y.; et al. Subsidence and Fusion Performance of a 3D-Printed Porous Interbody Cage with Stress-Optimized Body Lattice and Microporous Endplates - a Comprehensive Mechanical and Biological Analysis. Spine Journal 2022, 22, 1028–1037, doi:10.1016/j.spinee.2022.01.003.
  • Cuzzocrea, F.; Ivone, A.; Jannelli, E.; Fioruzzi, A.; Ferranti, E.; Vanelli, R.; Benazzo, F. PEEK versus Metal Cages in Posterior Lumbar Interbody Fusion: A Clinical and Radiological Comparative Study. Musculoskelet Surg 2019, 103, 237–241, doi:10.1007/s12306-018-0580-6.
  • Nemoto, O.; Asazuma, T.; Yato, Y.; Imabayashi, H.; Yasuoka, H.; Fujikawa, A. Comparison of Fusion Rates Following Transforaminal Lumbar Interbody Fusion Using Polyetheretherketone Cages or Titanium Cages with Transpedicular Instrumentation. European Spine Journal 2014, 23, 2150–2155, doi:10.1007/s00586-014-3466-9.
  • Tanida, S.; Fujibayashi, S.; Otsuki, B.; Masamoto, K.; Takahashi, Y.; Nakayama, T.; Matsuda, S. Vertebral Endplate Cyst as a Predictor of Nonunion after Lumbar Interbody Fusion: Comparison of Titanium And Polyetheretherketone Cages. Spine (Phila Pa 1976) 2016, 152, 28, doi:10.1097/BRS.0000000000001605.
  • Gok, M.G. Creation and Finite-Element Analysis of Multi-Lattice Structure Design in Hip Stem Implant to Reduce the Stress-Shielding Effect. Proceedings of the Institution of Mechanical Engineers, Part L: Journal of Materials: Design and Applications 2022, 236, 429–439, doi:10.1177/14644207211046200.

Judging from the current state of the art, what are the future development trends? What scientific methods will be more thoroughly studied in the future? These should be presented at the last.

Future works include the development of a repeatable and robust design methodology of biomedical implants with a combination of Gibson-Ashby model’s data, in silico and in vitro experiments. Data reported in the present review provide a scientific base for the choice of the optimal lattice topology and design parameters. (please see lines from 548 to 541)